# Comparing the Therapeutic Efficacies of Lung Cancer: Network Meta-Analysis Approaches

**DOI:** 10.3390/ijerph192114324

**Published:** 2022-11-02

**Authors:** Chuan-Hsin Chang, Yue-Cune Chang

**Affiliations:** 1Research Center for Chinese Herbal Medicine, Graduate Institute of Healthy Industry Technology, College of Human Ecology, Chang Gung University of Science and Technology, Taoyuan 33303, Taiwan; 2Department of Mathematics, Tamkang University, New Taipei City 25137, Taiwan

**Keywords:** air pollution, fine particulate matter (PM2.5), lung cancer, network meta-analysis

## Abstract

Background: In recent years, reduction of nuclear power generation and the use of coal-fired power for filling the power supply gap might have increased the risk of lung cancer. This study aims to explore the most effective treatment for different stages of lung cancer patients. Methods: We searched databases to investigate the treatment efficacy of lung cancer. The network meta-analysis was used to explore the top three effective therapeutic strategies among all collected treatment methodologies. Results: A total of 124 studies were collected from 115 articles with 171,757 participants in total. The results of network meta-analyses showed that the best top three treatments: (1) in response rate, for advanced lung cancer were Targeted + Targeted, Chemo + Immuno, and Targeted + Other Therapy with cumulative probabilities 82.9, 80.8, and 69.3%, respectively; for non-advanced lung cancer were Chemoradio + Targeted, Chemoradi + Immuno, and Chemoradio + Other Therapy with cumulative probabilities 69.0, 67.8, and 60.7%, respectively; (2) in disease-free control rate, for advanced lung cancer were Targeted + Others, Chemo + Immuno, and Targeted + Targeted Therapy with cumulative probabilities 93.4, 91.5, and 59.4%, respectively; for non-advanced lung cancer were Chemo + Surgery, Chemoradio + Targeted, and Surgery Therapy with cumulative probabilities 80.1, 71.5, and 43.1%, respectively. Conclusion: The therapeutic strategies with the best effectiveness will be different depending on the stage of lung cancer patients.

## 1. Introduction

Lung cancer is one of the most common malignancies worldwide and a leading cause of cancer-related death. The latest estimates on the global burden of cancer, which have been released by the International Agency for Research on Cancer (IARC), have reported that for lung cancer, there are an estimated 2.2 million new cancer cases and 1.8 million deaths, which is the second most commonly diagnosed cancer and the leading cause of cancer death, ranking third for incidence in 2020 [1]. Among them, lung cancer has been the leading cause of cancer death and more than 40% are non-smokers [2]. The previous studies have shown that there was a close relationship between air pollutants and human health, especially lung cancer [3].

The previous studies have shown that the increase of the use of coal-fired power plants has caused roughly seventy percent of the air pollution and emitted large quantities of various pollutants, including sulfur oxides (SOx), nitrogen oxides (NOx), carbon dioxide (CO_2_), ozone (O_3_), and volatile organic compound [4,5,6], as well as particulate matter (PM) with an aerodynamic diameter of less than 2.5 μm (PM2.5) [7,8]. The inhalable PM2.5, the tiny particles, can enter the bronchioles and terminal alveoli, stay in bronchial and alveolar cells for a long time, and eventually hamper macrophage function, resulting in enhancing pneumococcal infectivity and aggravating pulmonary pathogenesis [9]. Additionally, the studies show that this pollutant causes several health effects, including increased morbidity and mortality risk of asthma [10], central nervous system disease, cardiovascular problems [11,12,13,14,15], respiratory infections [16,17,18,19,20], lung cancer [7,21], and impaired lung development [21,22,23,24,25]. The studies also mentioned that the inhalation of PM2.5 will enter deeply into the lungs as well as the bloodstream and lodge in the heart, resulting in the formation of plaques, contributing to a series of hypertension symptoms [26,27]. The global proportion of lung cancer deaths attributing to outdoor ambient PM2.5 air pollution was 14% in 2017, ranging from 4.7% in the United States to 20.5% in China [28]. Hamra et al. showed the meta-relative risk for lung cancer associated with PM2.5 by continent was 1.09 (95% CI: 1.04, 1.14) [29]; the other meta-analysis in 2017 for lung cancer mortality associated with PM2.5 was highest in North America (1.15 (95% CI: 1.07, 1.24)), followed by Asia (1.12 (95% CI: 0.94, 1.35)), and then Europe (1.05 (95% CI: 1.01, 1.10)), and lung cancer incidence associated with PM2.5 is greatest in Asia (1.09 (95% CI: 1.03, 1.15)), followed by North America (1.06 (95% CI: 1.01, 1.11)), and then Europe (1.03 (95% CI: 0.61, 1.75)) [30]. It has been known that PM air pollution contributes to the incidence of lung adenocarcinoma in Europe, but the corresponding effect in East Asia is still uncertain [31]. In Asia, the highest air pollution-related health risk country is China, and the government of China has worked on the rapid improvement in air quality [32]. In Taiwan, air quality keeps dropping to unhealthy levels. Our previous longitudinal study shows that the exposure of PM2.5 is highly correlated with oxidative and methylated DNA damage in young adults [33]. Accordingly, either short- or long-term exposure of PM2.5 might cause severe damage in the human body (especially in young children and elderly people, as well as young adults).

Lung cancer is histologically divided into two main types: small cell lung cancer (SCLC) and non-small cell lung cancer (NSCLC) are 10–15% and 80–85%, respectively; NSCLC is further classified into three types: adenocarcinoma (the most common subtype of lung cancer), squamous cell carcinoma (the second most common subtype of lung cancer), and large cell carcinoma [34,35,36]. From the report of global surveillance of trends in cancer survival 2000–14 (CONCORD-3), the survival of patients with lung cancer at 5 years after diagnosis is only 10% to 20% in most countries [37]. Depending on the stage of NSCLC, patients are eligible for certain multimodal treatments, including surgery, radiation, chemotherapy, immunotherapy, and targeted therapy. The mainstay of treatment of early-stage NSCLC patients is surgery, and it has been shown that adjuvant chemotherapy has demonstrated providing an absolute survival benefit [38,39]. For NSCLC Stage III patients with unresectable disease, being treated with concurrent chemoradiation is the preferred treatment [40]; however, the results of the studies mention that most patients receiving consolidation chemotherapy after concurrent chemo-radiotherapy fail to provide survival benefit [41,42], and approximately only 15 to 30% of patients remain alive at 5 years [40,43]. Lately, several studies show that immunotherapy after concurrent chemoradiotherapy in NSCLC patients with stage III, who do not have disease progression [44,45,46,47,48]. Targeted/personalized medicine by targeting appropriate molecular targets, including epidermal growth factor receptor (EGFR), anaplastic lymphoma kinase (ALK), BRAF, vascular endothelial growth factor (VEGF), insulin-like growth factor-1 receptor (IGF-1R), and c-MET (MET), in tumors has been shown to improve survival in patients [49,50]. For the treatment of advance and metastatic lung cancer patients with EGFR-mutation and ALK-rearrangement, small-molecule EGFR tyrosine kinase inhibitors (TKIs) have been well-established and introduced in clinical practice in the last few years, including small-molecule EGFR-TKIs like erlotinib and gefitinib [51,52,53,54]. Due to low therapeutic efficacy of gefitinib, the use of gefitinib has been restricted in the USA [55], while it remains effective in Asia [56,57,58]. Since gefitinib has a good therapeutic effect on Asians, gefitinib have been under coverage by Taiwan’s National Health Insurance since 2007. Additionally, in addition to traditional cytotoxic chemotherapies, there are two primary classes of drugs for the treatment of systemic lung cancer, including many forms of small molecule-targeted therapies and immunotherapies [59]. It has been reported that few patients cured by EGFR-TKI alone eventually acquire resistance and relapse [60,61]. Accordingly, the discovery of next-generation EGFR-TKIs (second–third–fourth generation EGFR-TKIs) or development of the combined therapeutic strategy is required for improving the treatment effectiveness. In order to fill in the unmet medical needs, the efficient therapeutic strategies of lung cancer have been further investigated in order to stop tumor growth (even cancer disappeared), increase the survival rate, and prevent inducing grade 3 or 4 treatment-related toxicity, as well as improving life quality and controlling cancer-related pain.

In this study, we used meta-analysis and network meta-analysis to address the following study purposes: ignoring the effects of other possible prognostic factors, (1) the current used lung cancer treatments were effective in what efficacies indexes; (2) what are the most effective top three treatments?

## 2. Materials and Methods

### 2.1. Literature Search Strategy

The literature search was performed for the related research articles, mainly from the following electronic databases: PubMed, Cochrane Library, Google Scholar, and Airiti Library. To include as many lung cancer treatment-related articles as possible, we also searched a non-traditional journal, named Journal of Negative Results in BioMedicine. The search keywords we used were: (“Lung cancer” OR “Small cell lung cancer” OR “SCLC” OR “Non-small cell lung cancer” OR “NSCLC”) AND (Chemotherapy OR Radiotherapy OR EGFR* OR EGFR-TKI OR “Tyrosine Kinase Inhibitor” OR Surgery OR Platinum* OR “anaplastic lymphoma kinase” OR “ALK inhibitor” OR Crizotinib OR Ceritinib OR Carboplatin OR Erlotinib OR Gemcitabine OR Pemetrexed OR Bevacizumab OR Nivolumab OR Docetaxel OR Atezolizumab OR Gemcitabine OR Paclitaxel OR Thermotherapy OR “Shenqi Fuzheng Injection”) AND (“Quality of Life” OR QOL OR “Functional Assessment of Cancer Therapy-Lung” OR “FACT-L” OR “Lung Cancer Symptom Scale” OR “Progression-free Survival” OR PFS OR “Overall Survival” OR “Tumor Response” OR “Performance status” OR WBC OR Platelet OR PLT OR Hemoglobin OR HB OR Nausea OR Vomiting) AND (“Clinical Trial” OR “Clinical Study”). In addition, we also first searched for those meta-analyses or systematic reviews articles with the theme of lung cancer efficacy, and then found all the articles used in these articles. The final papers we included in this study were those satisfied the following inclusion and exclusion criteria. The search flow chart is presented in Figure 1.

### 2.2. Inclusion and Exclusion Criteria

Studies were included if they met the following inclusion criteria: (1) the participants have been diagnosed as the lung cancer; (2) the study provided at least one efficacy indicator (e.g., tumor response, disease-free survival rate, et al.); (3) the study provided sufficient information to evaluate the (Hedges’ g) effect size [35], or the authors responded to our mail and willing to provide additional information; (4) in addition to the efficacy index, there was at least one characteristic of participants’ information available, such as: average age, sex ratio, etc., or treatment-related information (such as: treatments, dose, evaluation scales, etc.).

Studies that met the following criteria were excluded: (1) the study did not provide sufficient information to evaluate the (Hedges’ g) effect sizes (e.g., lack of standard error), or the authors did not response to our mail or did not willing provide additional information; (2) the study was cross-sectional study; (3) the study did not meet the above inclusion criteria.

### 2.3. Data Extraction

The relevant information about the efficacy and its possible influencing factors we extracted from the collected articles were as follows: year of publication, mean age, and gender ratio. The scale used in these articles included smoking rate, pre-treated (yes vs. no), the severity of the disease (advanced vs. early-stage), and the treatment strategies. All efficacy-related indexes for comparisons were extracted and analyzed separately. When extracting the data of each efficacy index, there were, mainly, two types of data natures shown as follows: (1) for continuous variables, such as median survival time, disease-free survival time, 1, 3, 5 years survival rates, and 1, 3, 5 years disease-free survival rates, we extracted the value of each indicator and its corresponding standard error; (2) for binary category variables, such as tumor response (overall survival, disease-free survival, distance control survival), we extracted the number of events and non-events for each indicator in each group. If there were missing data caused by being unpresented in the collected articles, we contacted the authors to inquire about relevant information. If the authors did not response to our mail or were not willing to provide additional information, the article was excluded for further data analysis.

The analysis procedure of the statistical software we used is related to the names of treatments’ alphabetical order. Accordingly, it is necessary to recode the names of the treatments before doing the network meta-analysis. We recoded the treatment types as follows: (1) A_Surgery (Surgery); (2) B_Radio (Radiotherapy); (3) Chemo (Chemotherapy); (4) D_Target (Targeted Therapy); (5) E_ImmuT (Immunotherapy); (6) C1_Chemo + Surg (Chemotherapy + Surgery); (7) C2_ChemoRadio (Chemoradiotherapy); (8) C_Chemo (Chemo + Chemotherapy); (9) D3_Chemo + Target (Chemo + Targeted Therapy); (10) D4_Target + Target (Targeted + Targeted Therapy); (11) E3_Chemo + Immu (Chemo + Immunotherapy); (12) F2_Radio + Others (Radio + Others Therapy); (13) F3_Chemo + Others (Chemo + Others Therapy); (14) F4_Target + Others (Targeted + Others Therapy); (15) DC2_ChemoRadio + Target (ChemoRadio + Targeted Therapy); (16) EC2_ChemoRadio + Immu (ChemoRadio + Immunotherapy); (17) FC2_ChemoRadio + Others (ChemoRadio + Others Therapy).

### 2.4. Statistical Analysis

STATA/SE version V13.0 for Windows (Stata Corporation, College Station, TX, USA) was used for all statistical analyses. We first used the fixed effects meta-analysis to evaluate the pooled effects size. The random effects meta-analysis was followed if the testing results of between studies’ heterogeneity was highly significant. For binary outcome (the response rate and the disease-free control rate), we further used the network meta-analyses, trying to address the most critical question that gives rise to concern by both oncology doctors and lung cancer patients: after confirming the severity of lung cancer, what are the top three most effective treatments?

However, compared to other randomized placebo control clinical trials, the active control was a commonly used methodology for any stage of a lung cancer clinical trial due to the ethical issue. The commonly used effect sizes in the meta-analysis were the between groups difference of efficacies (whether it is in log scale, such as log (Odds Ratio), log (Rate Ratio), or log (Hazard Rate Ratio), or Hedges’ g unbiased standardized mean difference (SMD)). Therefore, the same treatment comparing with different active control groups, the final effect sizes obtained from these two different studies might end up with tremendous differences. Accordingly, the effect size we used in meta-analysis was the indicators of the efficacy of each treatment itself, such as: three years survival rate or overall response rate (RR), rather than the relative efficacy of the two treatments, such as hazard ratio or rate ratio. In this study, we tried to include as many related clinical trials as possible, and some single arm clinical trials (phase II) were also included in this study.

## 3. Results

There were 123 studies from 115 articles involving 171,757 patients in total with lung cancer who met the inclusion criteria after rigorous identification (Figure 1). The listing of the detail names of the first author and the titles are shown in Appendix A. The results were presented according to the pre-specified two study purposes and are shown as follows:

### 3.1. Results of Meta-Analyses

The indicators of treatment efficacies were presented according to disease severity, named advanced stage or early-stage. For the former, the commonly used indicators were median/mean survival time (in months), median/mean disease-free survival time, (overall) response rate, and disease-free control rate. For the latter, the commonly used indicators were 1, 3, 5 years survival rates and 1, 3, 5 years disease-free survival rates. All efficacy indicators (effect sizes) were evaluated in natural log scale (due to skew to the right distributed). The corresponding standard errors were obtained by delta-method. After that, the meta-analyses were done one indicator after another. The results were shown as follows.

#### 3.1.1. Advanced Stage

For each indicator, the random effect’s meta-analysis was used due to the highly significant heterogeneity chi-squared test (*p*-values < 0.001). The results were summarized and are shown in Table 1.

As shown in Table 1, for advance lung cancer patients: (1) the median/mean survival time and disease-free survival time were 13.46 and 5.22 months, respectively; response rate and disease-free control rate were 27.28% and 66.29%, respectively; (2) variations in effect sizes attributable to heterogeneity, I^2^, were 99.7%, 99.0%, 95.7%, and 96.0%, respectively; (3) estimates of between-study variance, τ^2^, were 0.7115, 0.4963, 0.3208, and 0.0578, respectively; (4) the number of available studies were 85, 95, 151, and 125, respectively.

#### 3.1.2. Early-Stage

The homogeneity chi-squared tests of six indicators of treatment efficacies, namely 1, 3, 5 years survival/disease-free survival rates, were all highly significant (all *p*-values < 0.001). The random effects of meta-analyses were used. The results are shown in Table 2.

For early-stage lung cancer patients, the results of random effects meta-analyses showed that: (1) as we expected, the survival rates and disease-free survival rates were decreased with respect to time; (2) variations in effect sizes attributable to heterogeneity, I^2^, were very high (greater than 95%); (3) estimates of between-study variance, τ^2^, were very low (lower than 0.2).

### 3.2. Results of Network Meta-Analyses

We further used network meta-analyses to address the following question: “Ignoring the effects of other possible influencing factors, what are the top three most effective treatment methods for response rate and disease-free survival (control) rate, respectively?”.

#### 3.2.1. The Results of Response Rate

The 17 treatment types we classified in the previous section almost included all possible types of current clinical trials designed for lung cancer treatment. However, due to some medical practical reasons, some current treatment methodologies were never put together for direct comparison. For example, the comparisons of treatment efficacies of Surgery vs. Chemotherapy or Surgery vs. Targeted therapy or Radiotherapy vs. Chemotherapy would never be options for designing a lung cancer clinical trial. Hence, when we put all these 17 treatment types into the network meta-analysis, the result of network map was not connected, and, as shown in Figure 2, it formed two disconnected components. In other words, an overall comparison among those 17 treatment types was not possible. According to Figure 2, we organized the network meta-analysis based on those two components for direct comparisons (similar to being analyzed by advanced stage and early-stage), and have presented them as follows:

#### 3.2.2. Advanced Stage (Component I)

We first conducted a network meta-analysis to those nine treatment types that can be compared directly, according to the component I in Figure 2 (Blue), and chose Chemo as the reference group. The result of the network map is shown in Figure 3. Among those nine treatments, we further used “Network rank max” and tried to figure out that “Ignoring the effects of other possible influencing factors, what are the top three most effective treatments among these 9 treatments?”. As shown in Table 3 and the rankogram (Figure 4), the best top three treatments for advanced lung cancer in response rate were as follows: (1) the chances of being the best treatment in the response rate of advanced lung cancer for Chemo + Immunotherapy (E3_Chemo + Immu), Targeted + Others Therapy (F4_Target + Others), and Targeted + Targeted Therapy (D4_Targeted + Targeted) were, respectively, 44.8%, 35.6%, and 17.0%; (2) the chances of being one of the best two treatments in the response rate of advanced lung cancer for Chemo + Immunotherapy (E3_Chemo + Immu), Targeted + Others Therapy (F4_Target + Others), and Targeted + Targeted Therapy (D4_Targeted + Targeted) were 69.3% (=44.8% + 24.5%), 58.0%, and 50.9%, respectively; (3) the chances of being one of the best three treatments in the response rate of advanced lung cancer for Targeted + Targeted Therapy (D4_Targeted + Targeted), Chemo + Immunotherapy (E3_Chemo + Immu), and Targeted + Others Therapy (F4_Targeted + Others) were 82.9% (=17% + 33.9% + 32%), 80.8%, and 69.3%, respectively.

#### 3.2.3. Non-Advanced Stage (Component II)

For the component II in Figure 2 (Gray) that chose A_Surgery as the reference group, we conducted another network meta-analysis for those eight treatment types that can be compared directly. The result of the network map is shown in Figure 5. Similarly, we further used “Network rank max” and tried to figure out that “Ignoring the effects of other possible influencing factors, what are the top three most effective treatments among these 8 treatments?”. As shown in Table 4 and the rankogram (Figure 6), the best top three treatments for non-advanced lung cancer in response rate were as follows: (1) the chance of being the best treatment in the response rate of non-advanced lung cancer for DC2_ChemoRadio + Target, FC2_ChemoRadio + Others, and F2_Radio + Others were, respectively, 33.3%, 25.4%, and 20.2%; (2) the chances of being one of the best two treatments in the response rate of non-advanced lung cancer for DC2_ChemoRadio + Target, FC2_ChemoRadio + Others, and EC2_ChemoRadio + Immu were 55.7% (=33.3% + 22.4%), 44.7%, and 41.3%, respectively; (3) the chances of being one of the best three treatments in the response rate of non-advanced lung cancer for DC2_ChemoRadio + Target, EC2_ChemoRadio + Immu, and FC2_ChemoRadio + Others were 69.0% (=33.3% + 22.4% + 13.3%), 67.8%, and 60.7%, respectively.

### 3.3. Disease-Free Control Rate

Similar to the response rate, the network map for disease-free control rate was disconnected (two components) and identical to Figure 2. Accordingly, we presented the network meta-analysis for disease-free control rate according to those two components for direct comparisons (similar to analyzed by advanced stage and early-stage), and showed as follows:

#### 3.3.1. Advanced Stage (Component I)

Chosing Chemo as the reference group, the result of network map was identical to Figure 3. Followed by “Network rank max” in the STATA, as shown in Table 5 and the rankogram (Figure 7), the best top three treatments for advanced lung cancer in disease-free control rate were as follows: (1) the chances of being the best treatment in the disease-free control rate of lung cancer for Targeted + Others Therapy (F4_Target + Others) and Chemo + Immunotherapy (E3_Chemo + Immu) were 78.1% and 20.0%, respectively; (2) the chances of being one of the best two treatments in the disease-free control rate of advanced lung cancer for Targeted + Others Therapy (F4_Target + Others), Chemo + Immunotherapy (E3_Chemo + Immu) and Targeted + Targeted Therapy (D4_Target + Target) were 90.3%, 61.8%, and 17.5%, respectively; (3) the chances of being one of the best three treatments in the disease-free control rate of advanced lung cancer for Targeted + Others Therapy (F4_Target + Others), Chemo + Immunotherapy (E3_Chemo + Immu), and Targeted + Targeted Therapy (D4_Target + Target) were 93.4%, 91.5%, and 59.4%, respectively.

#### 3.3.2. Non-Advanced Stage (Component II)

To rank the top three most effective treatments for non-advanced lung cancer in disease-free control rate among the eight treatments in component II, we chose A_Surgery as the reference group, and the result of the network map is shown in Figure 5. Followed by using “Network rank max”, as shown in Table 6 and the rankogram (Figure 8), the best top three treatments for non-advanced lung cancer in disease-free control rate are as follows: (1) the chances of being the best treatment in disease-free control rate of non-advanced lung cancer for DC2_ChemoRadio + Target, C1_Chemo + Surg and F2_Radio + Others were 33.3%, 22.5%, and 22.1%, respectively; (2) the chances of being one of the best two treatments in the disease-free control rate of non-advanced lung cancer for DC2_ChemoRadio + Target, C1_Chemo + Surg and F2_Radio + Others were 57.1%, 56.0%, and 32.0%, respectively; (3) the chances of being one of the best three treatments in the disease-free control rate of non-advanced lung cancer for C1_Chemo + Surg, DC2_ChemoRadio + Target, and A_Surgery were 80.1%, 71.5%, and 43.1%, respectively.

In summary, for advanced lung cancer, the top three treatments were Targeted + Others Therapy (F4_Target + Others), Chemo + Immunotherapy (E3_Chemo + Immu), and Targeted + Targeted Therapy (D4_Target + Target) in both response rate and disease-free control rate. However, for non-advanced lung cancer, the best top three treatments in response rate were DC2_ChemoRadio + Target, EC2_ChemoRadio + Immu, and FC2_ChemoRadio + Others. However, for disease-free control rate, the best top three treatments were C1_Chemo + Surg, DC2_ChemoRadio **+** Target, and A_Surgery.

## 4. Discussion

### 4.1. Summary of Findings

In this study, by using meta-analysis and network meta-analysis, we explored the efficacy of different treatment interventions in two different groups based on the severity of the disease (Advanced vs. early/non-advanced stage). The important findings in this study were as follows. First, the results of meta-regression suggested that, after adjusting for the effects of treatments and disease severity, the male to female ratio was negatively associated with all treatment efficacies (*p*-values range from 0.08 to <0.001, except for 1 year disease-free control rate). Second, the results show that the top three most effective treatment interventions for the advance stage lung cancer patients were: Chemo + Immunotherapy, Targeted + Others Therapy and Targeted + Targeted Therapy. In the non-advanced lung cancer patient group, the top three most effective treatment interventions were: ChemoRadio + Targeted Therapy, ChemoRadio + Others Therapy, and ChemoRadio + Immunotherapy.

### 4.2. Comparison with Previous Studies

#### 4.2.1. Sex Difference in Efficacy of Lung Cancer Treatment

In this study, the results of “male to female ratio were negatively associated with efficacies of all treatment interventions” provided indirect information that the treatment effects for male lung cancer patients were worse than female patients in general. According to the latest estimates on the global burden of cancer (released by IARC), lung cancer is the leading cause of cancer death in men in 93 countries [1], partially because of its high fatality rate [62]. The previously retrospective study reports that men, advanced or unresectable NSCLC patients, have a lower response rate to chemotherapy and longer survival than women [63]. Additionally, for efficacy of the combined treatment of chemotherapy and immunotherapy, the results of previous meta-analyses show that women with advanced lung cancer derived a statistically significantly larger benefit (overall survival hazard ratios) from the addition of chemotherapy to immunotherapy (anti-programmed cell death protein 1 (PD-1)/programmed cell death ligand 1 (PD-L1) as compared with men [64]. Additionally, it has been reported that males have a higher prevalence of sarcopenia, associated with worse treatment response and shorter long-term efficacy in NSCLC patients treated with immunotherapy, than females, which leads to worse PFS among male patients [65]. However, the other study shows opposite results—that immunotherapy improves overall survival for both male and female patients with advanced cancers (such as NSCLC), and men have a larger treatment effect from these drugs versus control treatments than women do [66]. Additionally, the other controversial results have reported that combined immunotherapy (PD-1/PD-L1 inhibitors) and chemotherapy significantly improve the overall survival and PFS in male patients, whereas combined treatment significantly benefits the overall survival but not the PFS in females [67]. The review study demonstrates that male patients harbor more uncommon EGFR mutations compared with common mutations among primary lung cancer patients in China, and patients with uncommon mutations might be unfavorable responses to EGFR-TKIs (the shorter PFS) compared with those with common mutations [68]. Accordingly, the sex difference of the treatment efficacy might be needed to further investigate.

#### 4.2.2. The Advance Stage Lung Cancer Patients

Chemotherapeutic agents without targeting specificity, in combination with rationally designed drugs (molecularly targeted drugs and immunotherapy drugs: checkpoint inhibitors) that selectively target cancer biomarkers, might improve the disease progression, and prevent cancer recurrence. According to the analysis results of this study, in the advance stage lung cancer patient group, the combined treatment of chemotherapy with immunotherapy was among the top three most effective treatment interventions for the advance stage lung cancer patients; the previous studies also show that the combination of immunotherapy (including Pembrolizumab and Atezolizumab) with chemotherapy (such as Platinum) or chemoradiotherapy is the most appropriate combined therapy for advanced NSCLC, which found the PFS and overall survival benefits halted the disease progression [44,45,46,69], with evaluating PD-L1 expression and/or tumor mutation burden (TMB) [70,71].

Additionally, the combined targeted therapy, including targeted agents (which were against EGFR, ALK, BRAF, IGF-1R, MET(c-MET), VEGF) and antibody–drug conjugate (ADC), as well as cancer vaccines, was among the top three most effective treatment interventions in this study. Regarding the treatment efficacy of the combined targeted therapy, the use of targeted therapy as a first-line treatment in combination with chemotherapy, as intercalated combination with chemotherapy, or as sequential therapy/maintenance therapy has been explored in many trials. The studies show that the advanced NSCLC patients receiving chemotherapy treatment (including pemetrexed, docetaxel, gemcitabine, cisplatin, and carboplatin) intercalated with targeted therapy (EGFR-TKI: erlotinib) has significantly prolonged PFS and is in favor of overall survival [72,73,74]. The data of randomized controlled trials mention targeted therapy (EGFR-TKI: erlotinib or gefitinib) as maintenance therapy improve objective RR and PFS, but this is unable to prolong overall survival [75]. However, as previously mentioned, the few patients treated with EGFR-TKI that had a poor response and acquired resistance and relapse [60,61] might be due to genotyping or clinical characteristics (including race, sex, histology, smoking status, pre-treatment, etc.). ALK mutations are approximately 3–7% of all lung tumors, and rearrangement in echinoderm microtubule-associated protein like 4-anaplastic lymphoma kinase (EML4-ALK) is the most common ALK rearrangement seen in NSCLC patients [76,77,78]. Currently, crizotinib is an FDA-approved agent that targets constitutively activated receptor tyrosine kinases resulting from EML4-ALK and other ALK-fusions. The data from the previous phase 3 trial study of ALK-positive metastatic NSCLC patients show that crizotinib significantly improves in PFS and RR compared with standard chemotherapy, and its safety profile was acceptable [79]. Besides, BRAF is a proto-oncogene, regulating signal transduction serine/threonine protein kinase, and is able to promote cell proliferation and survival [80]. BRAF mutations have been found in 1–4% of all NSCLC (1–2% of NSCLC patients are BRAF-V600E mutations [81]), most commonly in patients with adenocarcinomas [82,83,84,85]. Dual MAPK pathway inhibition using dabrafenib (BRAF inhibitor) plus trametinib (MEK inhibitor) achieved a 64% response rate and a median PFS of 10.9 months in BRAF-V600E mutation-positive NSCLC [86]; the data suggest that the combined treatment of BRAF targeted therapy and targeted agent against MEK in patients with BRAF mutation may be a useful therapeutic strategy in this subset of patients. Moreover, cancer vaccine: racotumomab-alum vaccine, an anti-idiotype vaccine targeting the N-glycolyl GM3 (NeuGcGM3) tumor-associated ganglioside, improves PFS and overall survival; accordingly, racotumomab-alum vaccine may be an effective and a well-tolerated treatment option for patients with advanced NSCLC [87].

#### 4.2.3. The Early-Stage/Non-Advanced Lung Cancer Patient

For the early-stage lung cancer patient group, the 5 year survival for patients with stage I NSCLC is around 80%, but only 13–60% patients with stage II to stage III disease have a 5 year survival [88]. The results of this study show that the combined chemoradiotherapy was the treatment interventions with the highest efficacy among the non-advanced lung cancer patients. Due to a lack of a reliable screening methods for detecting early-stage lung cancer, there are limited studies demonstrating treatment efficacy in early-stage lung cancer patients. The standard of care for patients with stage I [89,90] and II is surgical resection, and the addition of adjuvant chemotherapy in patients with stage I-II [39], stage IIIA, or selected stage IB offers a significant survival benefit [91]. However, it has been reported that nearly a third of patients with stage I NSCLC and at least 30% to 50% of patients with stage II and III NSCLC still die from recurrent disease treated with chemotherapy [92]. Besides, the previous study showed that patients with stage I to II small cell lung cancer treated with chemoradiotherapy have better outcomes compared with patients with stage III disease [93], while it has been reported that the chemotherapy combined with either surgery or plus radical radiotherapy gave a hazard ratio of 0.87 (13% reduction in the risk of death, equivalent to an absolute benefit of 5% at five years) or 0.87 (13% reduction in the risk of death; absolute benefit of 4% at two years) in all stages of NSCLC patients, respectively [94]. Recently, the use of neoadjuvant targeted therapy or immunotherapy in patients with NSCLC has been of particular interest.

The study shows that Adjuvant EGFR-TKI (gefitinib) leads to significantly longer disease-free survival with lower toxicity and better improvement of quality of life compared with that for vinorelbine plus cisplatin in Chinese patients with completely resected, stage II-IIIA, EGFR-mutant NSCLC [95]. Additionally, the data of ADAURA trial report that adjuvant EGFR-TKI (osimertinib) significantly improves in DFS in patients with stage IB/II/IIIA EGFR-mutant NSCLC [96]. The NCI-Canada (NCI-C) BR-19 study mentions that gefitinib does not improve overall survival with completely resected NSCLC without restricting to EGFR-TKI mutations [97]. As mentioned above, gefitinib shows effectiveness of therapeutic efficacy on Asians, but gefitinib has low therapeutic efficacy, restricted use, and is removed in both USA and Europe [55,98]. Regarding the development of first-, second-, and third-generations of EGFR-TKI, the controversy results of their treatment efficacy might be mainly due to ‘acquired resistance’ after EGFR-TKI treatment (either through secondary EGFR mutations or activation of EGFR-independent pathways) [99], genetic changes (including BRAF mutation [100,101], ALK-fusion [102], MET amplification/rearrangement/mutation [101], etc.), or the unexpected transformation into small cell lung cancer [103,104].

The review article of several clinical trials shows that adjuvant treatment with immunotherapy (also called immune checkpoint inhibitors; ICIs) after chemotherapy improves disease-free survival and may play a critical role in reducing disease recurrence in early-stage NSCLC patients [105]. Additionally, several studies report radiation-induced immunomodulatory effects in the local tumor microenvironment (including rendering tumor cells more susceptible to T-cell-mediated attack, enhancing expression of MHC class I as well as cell-adhesion molecules/other immunomodulatory-PD-L1-1, upregulating the release of chemokines) [106,107,108,109,110], which support a synergistic combination approach with immunotherapy to provide the improved clinical benefit [111].

### 4.3. Strengths and Limitations

In this study, we used network meta-analyses to figured out, for both advanced stage and non-advanced stage lung cancer, the top three most effective treatment methods for response rate and disease-free survival (control) rate, respectively, ignoring the effects of other possible influencing factors. However, the treatment efficacy could be influenced by some prognostic factors, for example: patient’s age, gender, severity of the disease (stage), etc. Accordingly, how to enhance the current network meta-analysis to be able to adjust for the effects of confounding variables simultaneously is heavily needed.

## 5. Conclusions

In summary, the current study showed the evidence-based comparative efficacy of the treatment strategies for lung cancer patients. We used the network meta-analyses to elucidate the top three most effective treatment interventions, for the response rate and disease-free survival (control) rate, of the lung cancer patients in advanced/non-advanced stages. Recently, magnetic nanoparticles have become a promising approach for either enhancing cancer diagnosis or as novel cancer therapeutic agents, along with conventional anticancer drugs or radiotherapy [112,113]. However, those novel therapeutic strategies might need to further evaluate their therapeutic efficacies, as well as the cause of adverse effects. In the future, by using network meta-analyses, we might have a chance to rearrange the top three most effective therapies for different stages of lung cancer patients with/without adjusting for the effects of some potential confounding variables.

## Figures and Tables

**Figure 1 ijerph-19-14324-f001:**
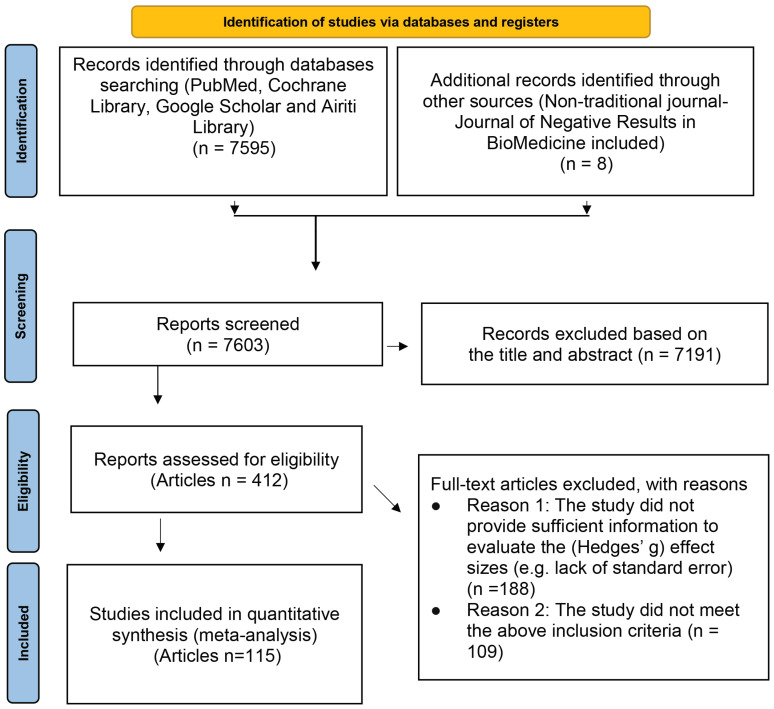
PRISMA flow diagram.

**Figure 2 ijerph-19-14324-f002:**
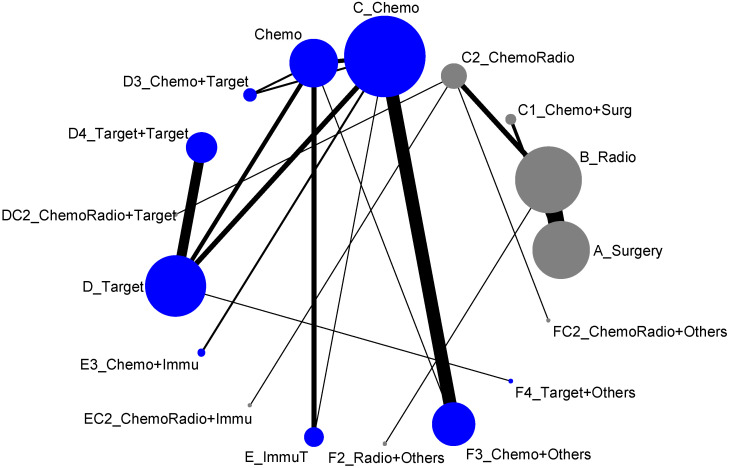
Network map of response rate/disease-free control (all treatment groups), blue is component I, and gray is component II.

**Figure 3 ijerph-19-14324-f003:**
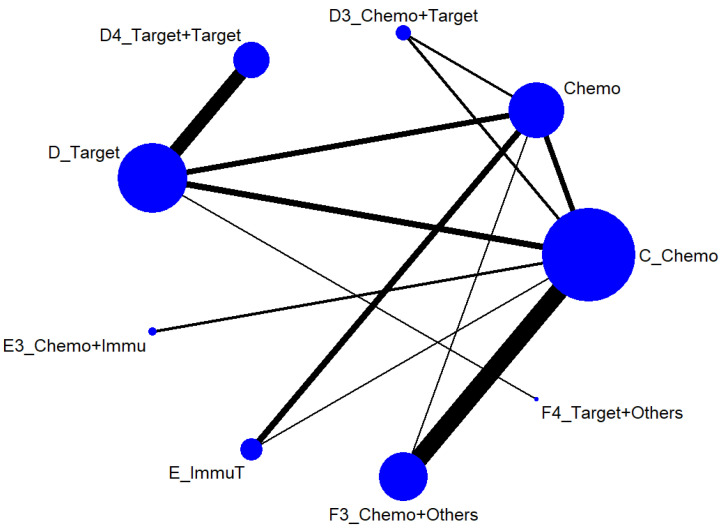
Network map of response rate/disease-free control rate (advanced stage).

**Figure 4 ijerph-19-14324-f004:**
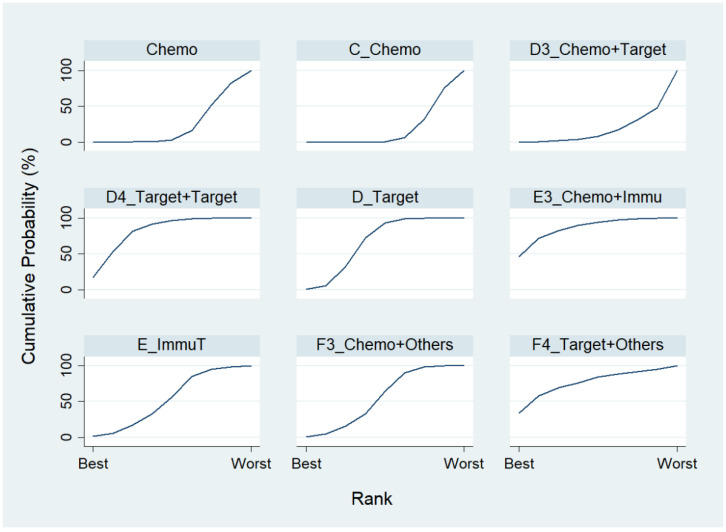
Rankogram for the advanced lung cancer in response rate.

**Figure 5 ijerph-19-14324-f005:**
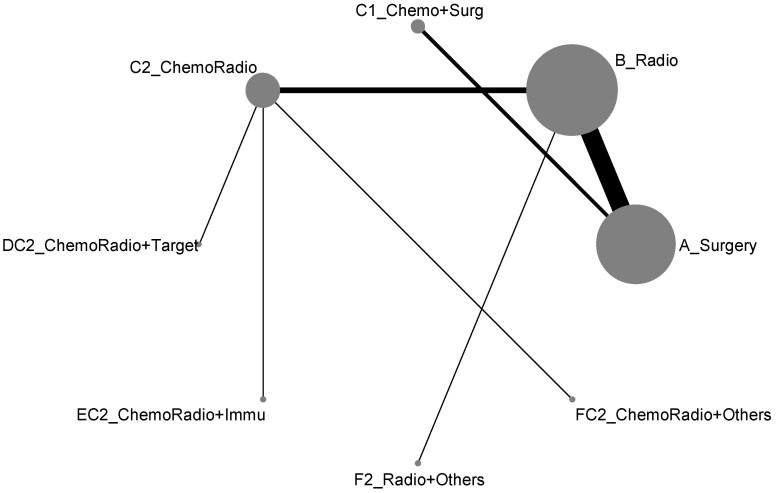
Network map of response rate/disease-free control rate (non-advanced stage).

**Figure 6 ijerph-19-14324-f006:**
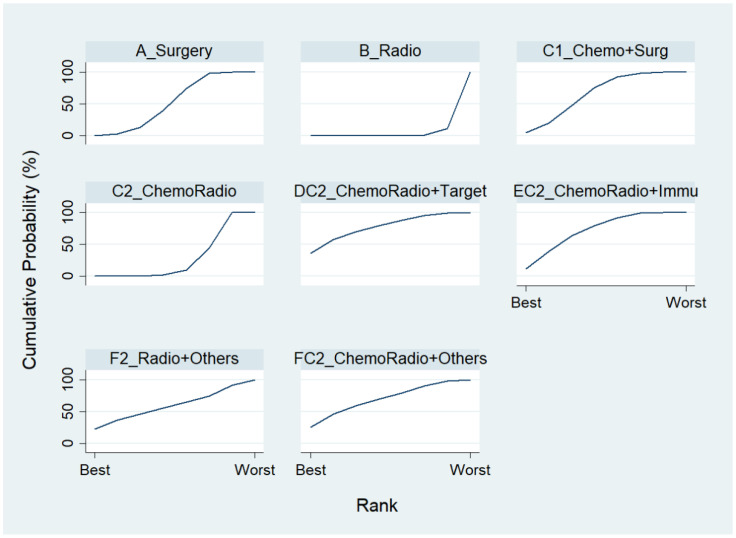
Rankogram for the non-advanced lung cancer in response rate.

**Figure 7 ijerph-19-14324-f007:**
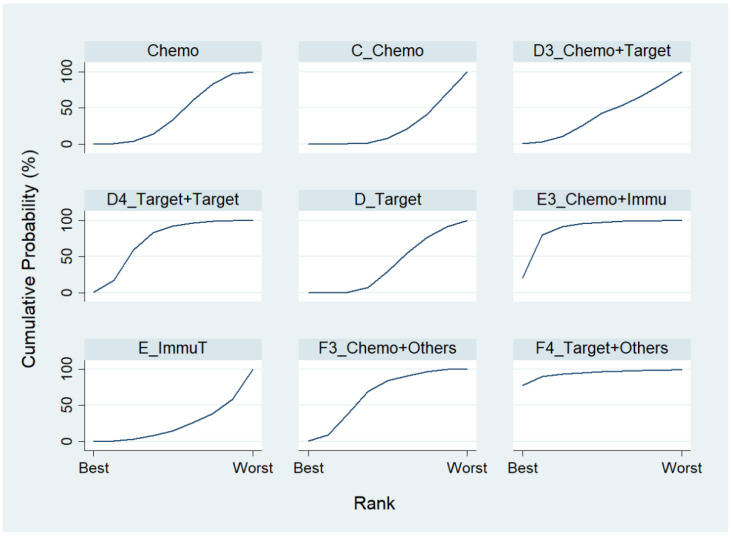
Rankogram for advanced lung cancer in disease-free control rate.

**Figure 8 ijerph-19-14324-f008:**
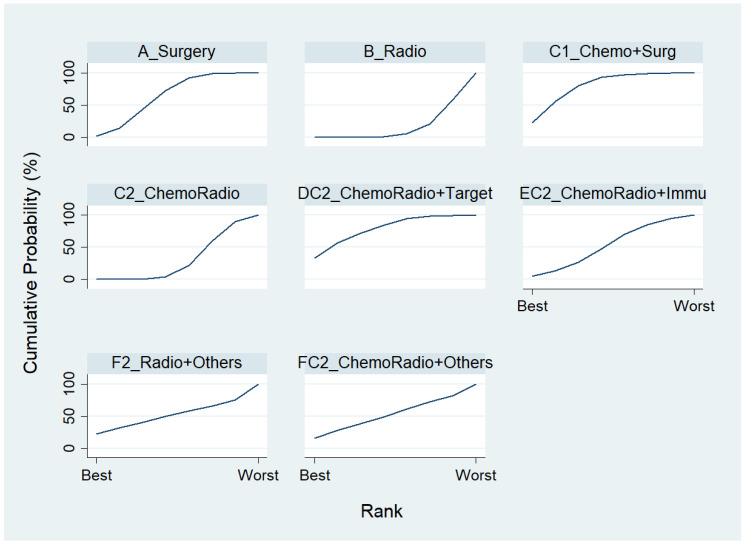
Rankogram for the non-advanced lung cancer in disease-free control rate.

**Table 1 ijerph-19-14324-t001:** Results of random effects meta-analyses for median/mean survival time (in months), median/mean disease-free survival time, response rate, and disease-free control rate.

	Pooled	95% C.I.	I^2^	τ^2^	n
Median/Mean Survival Time	13.46	(11.21, 16.17)	99.70%	0.7115	85
Median/Mean Disease-free Survival Time	5.22	(4.51, 6.03)	99.00%	0.4963	95
Response Rate	27.28	(24.73, 30.05)	95.70%	0.3208	151
Disease-free Control rate	66.29	(63.37, 69.34)	96.00%	0.0578	125

**Table 2 ijerph-19-14324-t002:** Results of random effects meta-analyses for 1, 3, 5 years survival rates and 1, 3, 5 years disease-free survival rates.

	Pooled	95% C.I.	I^2^	τ^2^	n
1 Year Survival rate	0.69	(0.68, 0.71)	97.80%	0.0125	123
3 Years Survival rate	0.44	(0.41, 0.47)	99.00%	0.0995	85
5 Years Survival rate	0.36	(0.31, 0.40)	99.30%	0.1909	53
1 Year Disease-free survival rate	0.48	(0.45, 0.51)	97.80%	0.0897	89
3 Years Disease-free survival rate	0.39	(0.35, 0.43)	97.70%	0.1514	57
5 Years Disease-free survival rate	0.37	(0.33, 0.42)	95.70%	0.1292	39

**Table 3 ijerph-19-14324-t003:** Estimated cumulative probabilities (%) of each treatment being the best in response rate (Advanced stage, Component I).

Rank	Chemo	C_Chemo	D3_Chemo + Target	D4_Target + Target	D_Target	E3_Chemo + Immu	E_ImmuT	F3_Chemo + Others	F4_Target + Others
**Best**	0	0	0	**17**	0.3	**44.8**	1.6	0.7	**35.6**
**2nd**	0	0	0.5	**33.9**	7.8	**24.5**	5.8	5.1	**22.4**
**3rd**	0.1	0.1	1.3	**32**	24.2	**11.5**	9.5	10	**11.3**
4th	0.2	0	1.9	9.5	43.5	8.1	13.1	15.4	8.3
5th	2.2	0.7	4.5	4.7	17.3	5.8	24.5	33.8	6.5
6th	10.3	7.4	10.2	1.9	6	3.4	29.3	25.5	6
7th	39.7	23.6	13	0.5	0.9	1.3	9.8	7.4	3.8
8th	31.4	44	16	0.3	0	0.4	4.2	1.8	1.9
Worst	16.1	24.2	52.6	0.2	0	0.2	2.2	0.3	4.2

**Table 4 ijerph-19-14324-t004:** Estimated cumulative probabilities (%) of each treatment being the best in response rate (Non-advanced stage, Component II).

Rank	A_Surgery	B_Radio	C1_Chemo + Surg	C2_ChemoRadio	DC2_ChemoRadio + Targe	EC2_ChemoRadio + Immu	F2_Radio + Others	FC2_ChemoRadio + Other
**Best**	0.3	0	5.7	0	**33.3**	**15.1**	**20.2**	**25.4**
**2nd**	2.8	0	15	0	**22.4**	**26.2**	14.3	**19.3**
**3rd**	10.6	0	23.3	0	**13.3**	**26.5**	10.3	**16**
4th	26.6	0	28.3	1.5	9.3	13.4	11.2	9.7
5th	37.4	0	18.5	7.6	7.5	10	9.2	9.8
6th	20.5	0.2	7.4	35.3	7	8.1	9.8	11.7
7th	1.8	10.7	1.8	55.5	6	0.7	17.4	6.1
Worst	0	89.1	0	0.1	1.2	0	7.6	2

**Table 5 ijerph-19-14324-t005:** Estimated cumulative probabilities (%) of each treatment being the best in disease-free control rate (Advanced Stage, Component I).

Rank	Chemo	C_Chemo	D3_Chemo + Target	D4_Target + Target	D_Target	E3_Chemo + Immu	E_ImmuT	F3_Chemo + Others	F4_Target + Others
**Best**	0	0	0.6	**0.7**	0	**20**	0	0.6	**78.1**
**2nd**	0.3	0	2.6	**16.8**	0	**59.8**	0.4	7.9	**12.2**
**3rd**	3.5	0.1	6.9	**41.9**	0.9	**11.7**	2.5	29.4	**3.1**
4th	9.9	1.2	15.4	24.1	6.6	4.6	4.9	31.2	2.1
5th	20.1	6.4	17.5	9.2	22.3	1.4	6.9	15.1	1.1
6th	26.9	13.2	10.4	3.9	25.2	1.3	11.3	7	0.8
7th	22.4	20.2	13.7	2.2	21.6	0.7	13.1	5.4	0.7
8th	14	30	15.3	1	15.5	0.4	19.9	3.4	0.5
Worst	2.9	28.9	17.6	0.2	7.9	0.1	41	0	1.4

**Table 6 ijerph-19-14324-t006:** Estimated cumulative probabilities (%) of each treatment being the best in disease-free control rate (Non-advanced stage, Component II).

Rank	A_Surgery	B_Radio	C1_Chemo + Surg	C2_ChemoRadio	DC2_ChemoRadio + Targe	EC2_ChemoRadio + Immu	F2_Radio + Others	FC2_ChemoRadio + Other
**Best**	**1.5**	0	**22.5**	0	**33.3**	4.4	**22.1**	16.2
**2nd**	**12.1**	0	**33.5**	0	**23.8**	9.2	9.9	11.5
**3rd**	**29.5**	0.1	**24.1**	0.2	**14.4**	13.3	8.1	10.3
4th	29.1	1	13	3.4	13.1	20.2	9.6	10.6
5th	20.3	4.1	3.6	18.4	10	22.8	8.3	12.5
6th	6.3	15.3	2.8	37.7	4.1	15.2	7.6	11
7th	1.2	37.8	0.5	30.4	0.9	9.3	9.9	10
Worst	0	41.7	0	9.9	0.4	5.6	24.5	17.9

## Data Availability

The datasets analyzed during meta-analysis and network meta-analysis are available from the corresponding author on reasonable request.

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
