# Peer review of "Comparing the Therapeutic Efficacies of Lung Cancer: Network Meta-Analysis Approaches"

_ijerph, 2022, doi:10.3390/ijerph192114324_

Round 1

Reviewer 1 Report

In this manuscript Chang et. al. have compared the efficacies of various therapeutics used in lung cancer using meta-analysis.

Here are my following concerns with this manuscript:

a) There are many manuscripts addressing drug efficacies against lung cancer using meta-analysis. The manuscript fails to clearly explain how this is an original study or what is unique about it that has never been analyzed before. They have left out a critical study that was published last year from their literature survey PMID: 34682798  and I'm not sure how this paper adds to the information that is already published. If you would have compared the world efficacy data with Taiwan data then I believe it will be original or else I'm not sure what is the originality of the paper.

b) There is strong disconnect between the introduction and the results. They have mentioned two major forms of cancer but the results are not connected to it. If one is not associating the drug treatment with a particular type of cancer is an awful waste of the study and effort.

c) Another disconnect is discussing Taiwan. They analyzed the worldwide data but did not compare how the treatment given in Taiwan and patient response compares to the rest of the world. If you are not planning to do that then why mention Taiwan in the first place. I also wonder if you mention it then what additional information will this manuscript provide compared to  PMID: 34682798. 

d) I did not understand in figure 1 where after removing duplicate data the number of records increase more than the original numbers they initially started the study with. Something is seriously wrong here.

e) For all tables and data, you need to differentiate between the cancer types, stages if possible and whether they are Taiwan or other parts of the world.

f) No effort was put in to explain the Rankogram. They are not obvious and self-explainatory.

g) How is table 3 different from table 5 and how is table 4 different from table 6?

h) Again the result outcomes from the analysis have not been clearly explained throughout the manuscript.

i) The discussion ends abruptly and it does not put the results in perspective to what is already known and how it compares. Is the outcome related to the whole world? If so, then remove the Taiwan context totally from your manuscript as you are not conducting any analysis specific to Taiwan.

Overall, there are many typos and the manuscript requires a major rewriting before it is accepted.

Author Response

We thank the reviewer for their careful reading of the manuscript and their constructive remarks. We have taken the comments on board to improve and clarify the manuscript. Please find below a detailed point-by-point response to all comments (reviewers’ comments in black, our replies in blue).

Here are my following concerns with this manuscript:

  1. a) There are many manuscripts addressing drug efficacies against lung cancer using meta-analysis. The manuscript fails to clearly explain how this is an original study or what is unique about it that has never been analyzed before. They have left out a critical study that was published last year from their literature survey PMID: 34682798 and I'm not sure how this paper adds to the information that is already published. If you would have compared the world efficacy data with Taiwan data then I believe it will be original or else I'm not sure what is the originality of the paper.

Ans: The literature search in this study was from world wide web and database. We used network meta-analysis to combine evidence on multiple randomized trials ranking the efficacies of all collected treatments. The major feature of network meta-analysis is the ability to combine direct and indirect evidence. More specifically, the comparison of treatments A and B could be obtained from either using studies that directly compare A with B (direct evidence) or using studies that compare A with C and B with C (indirect evidence). Differences from PMID: 34682798 to this submitted manuscript are as following: (1) this manuscript focused on ranking the efficacies of different therapeutic strategies for lung cancer in worldwide instead of Taiwan alone; (2) the data of PMID: 34682798 was collected from the Taiwan Cancer Registry (from 2010 to 2016). It was not a clinical trial. So, the results in PMID: 34682798 could not be included in this study.

  1. b) There is strong disconnect between the introduction and the results. They have mentioned two major forms of cancer but the results are not connected to it. If one is not associating the drug treatment with a particular type of cancer is an awful waste of the study and effort.

Ans: Yes, histologically, the primary lung cancer is classified (by the type of cells in which the cancer starts growing) as NSCLC (80~85%) and SCLC (10~15%). Using search keywords in this study, we included 115 papers and there was only one SCLC paper (No. 27, check the S1 Table). We also found that many papers focus on comparing the treatment effects of “advanced NSCLC”, “previously treated advanced NSCLC”, “early stage NSCLC”, or “Stage IB to IIIA NSCLC” (the last two were non-advanced stage) and highlighted in the titles, see S1 Table. Accordingly, we used network meta-analysis to identify the top three of the most effective therapeutic strategies according to advanced and non-advanced stages which is also a suggestion based on the results (there were two disconnected components) from the network meta-analysis (Figure 2).

  1. c) Another disconnect is discussing Taiwan. They analyzed the worldwide data but did not compare how the treatment given in Taiwan and patient response compares to the rest of the world. If you are not planning to do that then why mention Taiwan in the first place. I also wonder if you mention it then what additional information will this manuscript provide compared to PMID: 34682798.

Ans: Thank you for your careful review for this manuscript. In the introduction section, we mentioned Taiwan current energy policy would dramatically increase the risk of lung cancer which motivated to conduct this study. As a public health researcher, we conjecture people live in Taiwan would like to know “after confirming the severity of lung cancer, what are the top three most effective treatments?”. We apology for causing this kind of confusing. This manuscript focuses on worldwide data instead of Taiwan alone. Accordingly, we already modified our description in the revised manuscript. 

  1. d) I did not understand in figure 1 where after removing duplicate data the number of records increase more than the original numbers they initially started the study with. Something is seriously wrong here.

Ans: We apologize for happening this confusion. The truth is that we excluded those duplication during the searching procedure. An updated Figure 1 is used in the revised manuscript.

  1. e) For all tables and data, you need to differentiate between the cancer types, stages if possible and they are Taiwan or other parts of the world.

Ans: Yes, for treatment effects, these two factors are potential confounding variables. Their impacts to treatment effects needed to be adjusted for (subgroups analysis is one way to do it). However, this study is a meta-analysis which synthesizes summaries and conclusions from published papers. As you can see in the search keywords, we used both “SCLC” and “NSCLC” equally weighted. However, within 115 included papers, there was only one SCLC paper (No. 27, check the S1 Table). One of the possible reasons is that there were more than 80% lung cancers were NSCLC. That was the reason why we did not present the results in the cancer types.

On the other hands, the reasons we present the results in advanced and non-advanced stages instead of all stages were: (1) many papers focus on comparing the treatment effects of “advanced NSCLC”, “previously treated advanced NSCLC”, “early stage NSCLC”, or “Stage IB to IIIA NSCLC” (the last two were non-advanced stage) and highlighted in the titles, see S1 Table; (2) a suggestion based on the results (there were two disconnected components) from the network meta-analysis (Figure 2).

The reason we did not presented the results to differentiate between Taiwan or other parts of the world was that we focused on ranking the efficacies of different therapeutic strategies for lung cancer in worldwide instead of Taiwan alone.

  1. f) No effort was put in to explain the Rankogram. They are not obvious and self-explanatory.

Ans: We thanks for you to point this out. In the revised manuscript, we added the values of the cumulative probability into the interpretation of table 3, 4, 5, and 6. The y-axis of the corresponding Rankogram is the cumulative probabilities and the x-axis is the “Rank” in the first column of the table, which is: Best, 2nd, 3rd, 4th, 5th, 6th, 7th, and Worst.

  1. g) How is table 3 different from table 5 and how is table 4 different from table 6?

Ans: Table 3 and Table 5 are the top three treatments (in terms of the best, 2nd, and 3rd cumulative probabilities), in response rate and disease-free control rate, respectively, for advanced stage lung cancer. Table 4 and Table 6 are the similar results for non-advanced stage lung cancer. As we can see that the results in Table 3 and 5, for the advanced stage, the top three treatments remain the same in both response rate and disease-free control rate. However, for non-advanced stage the results are different. We added a summary of the above results in the end of results section of the revised manuscript (line 350~356).

  1. h) Again, the result outcomes from the analysis have not been clearly explained throughout the manuscript.

Ans: Thank you for your suggestion, we already revised results section and summarized the main findings in this study in Discussion section. In briefly, by using meta-analysis and network meta-analysis, we defined the effective treatments of lung cancer patients with different stages. Interestingly, the data in this study also showed that the treatment effectiveness was different between male and female, which might need to be further investigated.

  1. i) The discussion ends abruptly and it does not put the results in perspective to what is already known and how it compares. Is the outcome related to the whole world? If so, then remove the Taiwan context totally from your manuscript as you are not conducting any analysis specific to Taiwan.

Ans: Thank you for your suggestion, we already removed “the Taiwan context” and reorganized the sentences in this manuscript.

Overall, there are many typos and the manuscript requires a major rewriting before it is accepted.

Reviewer 2 Report

The manuscript titled “Comparing the Therapeutic efficiencies of Lung Cancer: Network Meta-analysis approaches” by Chang, C-H.; and Chang, Y-C.; is an original work where the authors established a network meta-analysis to deeply study cases of lung carcinoma in different maturation states and the types of therapies used in each state. More than 170,000 patients were taken into account which is a sign of the robustness of the test conducted by the authors. The conducted research is highly innovative combining the above described approaches and could pave the way in the production of the next-generation biocomputational analysis focused on monitoring human malignancies. The gathered findings may be relevant for the examined field. The results achieved are well-discussed during the main body of the reported manuscript. The scientific paper is well written. In my opinion the present manuscript is innovative and the methodological approached used matches with the scope of International Journal of Environmental Research and Public Health. For the above described reasons, I recommend the publication in International Journal of Environmental Research and Public Health once the following remarks will be fixed:

--------

INTRODUCTION

Introduction section is clear and concise. Furthermore, this section provides accurate information with relevant references of the system of study. Authors should pay attention in the following points:

I)       “In Taiwan, it was estimated around 51,0161 deaths from cancer occurred in 2019 [2]” (lines 34-35). Please, the authors should take care about the figure of 51,0161.

II)    “(…) large quantities of various pollutants, including  sulfur oxides (SOx), nitrogen oxides (NOx), carbon dioxide (CO2), ozone (O3), and volatile compounds [9-11]” (lines 48-50). Could the authors quantify the emissions of these pollutants? Moreover, the “2” and “3” from CO2 and O3 should be indicated as subscript.

III) “(…) improvement life quality and the control of cancer related pain” (line 115). Authors should fix the issue regarding the strikethrough words.

--------

MATERIALS AND METHODS

Authors have fully explained all the required information conducted in this scientific work. No actions are required.

--------

RESULTS

The most significant outcomes are perfectly explained for all potential target audiences. Nevertheless, the following points should be addressed:

I)          “For latter, the commonly used indicators were 1, 3, 5 years survival rates and 1, 3, 5 years disease-free survival rates (lines 217-218). Why did the authors not consider disease-free/survival rates higher than 5 years? Table 2 (from line 240) should be checked on this regard.

II)       Figure 2 (line 265). I consider interesting the addition of percentage values for each type of cancer treatment beside their respective therapy. Same comment for Figure 3 (line 288) and Figure 5 (line 309).

--------

DISCUSSION

Authors perfectly state the most important outcomes gathered in this submitted manuscript.

“4.2.1. Sex difference in efficacy of lung cancer treatment” (line 372). Authors studied the underlying response of lung cancer treatments according the gender. Did the authors conduct the same research regarding the age of the patients?

“Strengths and Limitations” (lines 493-510). Here, it may be convenient if the authors present other scenarios where their employed methodology could serve to discern still remaining open questions.

--------

CONCLUSIONS

Conclusion section should be opened up to potential future perspectives by adding a small statement.  The methodology presented by the authors could perfectly complement to recently developed strategies to fight against cancer diseases leading a better prognosis of how efficient these therapies are. In this context, novel magnetic nanoparticles with rod-shape morphologies [1] and after being coated by organic polymers to increase their biocompatibility and thus, serving as drug-delivery to specific target tumor locations [2].

[1] Marcuello, C.; Chambel, L.; Rodrigues, M.S.; Ferreira, L.P.; Cruz, M.M. Magnetotactic Bacteria: Magnetism Beyond Magnetosomes. IEEE Trans. Nanobioscience. 2018, 17, 555-559. https://doi.org/10.1109/TNB.2018.2878085.

[2] Alromi, D.A.; Madani, S.Y.; Seifalian, A. Emerging Application of Magnetic Nanoparticles for Diagnosis and Treatment of Cancer. Polymers 2021, 13, 4146. https://doi.org/10.3390/polym13234146.

--------

REFERENCES

Bibliography citations are in the proper format of International Journal of Environmental Research and Public Health.

--------

OVERVIEW AND FINAL COMMENTS

The submitted work is well-designed and the gathered results are interesting for the evaluation of treatment response in patients not only in cancer but also expandable for other human diseases. For this reason, I will recommend the present scientific manuscript for further publication in International Journal of Environmental Research and Public Health once all the aforementioned suggestions will be properly fixed.

Author Response

We thank the reviewer for their careful reading of the manuscript and their constructive remarks. We have taken the comments on board to improve and clarify the manuscript. Please find below a detailed point-by-point response to all comments (reviewers’ comments in black, our replies in blue).

INTRODUCTION

Introduction section is clear and concise. Furthermore, this section provides accurate information with relevant references of the system of study. Authors should pay attention in the following points:

  1. I) “In Taiwan, it was estimated around 51,0161 deaths from cancer occurred in 2019 [2]” (lines 34-35). Please, the authors should take care about the figure of 51,0161.

Ans: We thanks for your kindly reminded this typo. It should be 50,161. Although the updated precise number was 50,232, we decided to remove all Taiwan related context from this revised manuscript, due to the following reasons: 1. The literature search in this study was from world wide web and database; 2. We focused on ranking the efficacies of different therapeutic strategies for lung cancer in worldwide instead of Taiwan alone; 3. A suggestion from other reviewer. Therefore, we reorganized the contexts in introduction section, and the line number changes.

  1. II) “(…) large quantities of various pollutants, including sulfur oxides (SOx), nitrogen oxides (NOx), carbon dioxide (CO2), ozone (O3), and volatile compounds [9-11]” (lines 48-50). Could the authors quantify the emissions of these pollutants? Moreover, the “2” and “3” from CO2 and O3 should be indicated as subscript.

Ans: Thank you for your suggestion, we already made revision. The line number of this manuscript has been changed. For the emissions of the pollutants, those might be various in different cities, countries, and time. However, the real time air quality index of Taiwan is available from the website: https://airtw.epa.gov.tw/ENG/default.aspx .

III) “(…) improvement life quality and the control of cancer related pain” (line 115). Authors should fix the issue regarding the strikethrough words.

Ans: We thanks for your kindly remind and remove the strikethrough words in the revised manuscript.

--------

MATERIALS AND METHODS

Authors have fully explained all the required information conducted in this scientific work. No actions are required.

--------

RESULTS

The most significant outcomes are perfectly explained for all potential target audiences. Nevertheless, the following points should be addressed:

  1. I) “For latter, the commonly used indicators were 1, 3, 5 years survival rates and 1, 3, 5 years disease-free survival rates (lines 217-218). Why did the authors not consider disease-free/survival rates higher than 5 years? Table 2 (from line 240) should be checked on this regard.

Ans: This is an interesting question. We both personally would like to know that too. However, network meta-analysis is a meta-analysis that synthesize summaries and conclusions from the published literatures. For a clinical silence disease like lung cancer, most published papers only presented 5 years survival/disease-free control rates for early stage (or stage I B ~ IIIA) and less than 24 months survival/disease-free control rates for advance stage.

  1. II) Figure 2 (line 265). I consider interesting the addition of percentage values for each type of cancer treatment beside their respective therapy. Same comment for Figure 3 (line 288) and Figure 5 (line 309).

Ans: You ARE an excellent expert in this area. Yes, that is an interesting problem. However, that information is not available from current network meta-analysis output. Obviously, there are a lot of space for improvement in the network meta-analysis.

--------

DISCUSSION

Authors perfectly state the most important outcomes gathered in this submitted manuscript.

“4.2.1. Sex difference in efficacy of lung cancer treatment” (line 372). Authors studied the underlying response of lung cancer treatments according the gender. Did the authors conduct the same research regarding the age of the patients?

Ans: Yes. In this study, we used meta-regression to explore the impact of age, gender, and pre-treated patient on median/mean overall survival/disease free survival time, and response/disease free control rate for advanced stage. And, for early stage, we explored their impact on 1, 3, and 5 years survival/disease free control rate. However, to be focus on ranking treatment efficacies, we decided to exclude the results of meta-regression. However, in discussion section, we compared the results with previous studies which we thought some readers might interested in it.

“Strengths and Limitations” (lines 493-510). Here, it may be convenient if the authors present other scenarios where their employed methodology could serve to discern still remaining open questions.

Ans: We appreciate your helpful suggestions. We rewrite this section from lines 494-500.

CONCLUSIONS

Conclusion section should be opened up to potential future perspectives by adding a small statement.  The methodology presented by the authors could perfectly complement to recently developed strategies to fight against cancer diseases leading a better prognosis of how efficient these therapies are. In this context, novel magnetic nanoparticles with rod-shape morphologies [1] and after being coated by organic polymers to increase their biocompatibility and thus, serving as drug-delivery to specific target tumor locations [2].

Ans: Thank you for your great suggestion; we added the novel and potential therapeutic approaches in conclusion sections (lines 502-512). However, the effectiveness of those and their safety profile might need to be further investigated. Also, meta-analyses require sufficient data; in the future, we plan to include those new cancer therapies to explore their therapeutic effectiveness for lung cancer patients. 

[1] Marcuello, C.; Chambel, L.; Rodrigues, M.S.; Ferreira, L.P.; Cruz, M.M. Magnetotactic Bacteria: Magnetism Beyond Magnetosomes. IEEE Trans. Nanobioscience. 2018, 17, 555-559. https://doi.org/10.1109/TNB.2018.2878085.

[2] Alromi, D.A.; Madani, S.Y.; Seifalian, A. Emerging Application of Magnetic Nanoparticles for Diagnosis and Treatment of Cancer. Polymers 2021, 13, 4146. https://doi.org/10.3390/polym13234146.

--------

REFERENCES

Bibliography citations are in the proper format of International Journal of Environmental Research and Public Health.

--------

OVERVIEW AND FINAL COMMENTS

The submitted work is well-designed and the gathered results are interesting for the evaluation of treatment response in patients not only in cancer but also expandable for other human diseases. For this reason, I will recommend the present scientific manuscript for further publication in International Journal of Environmental Research and Public Health once all the aforementioned suggestions will be properly fixed.

Reviewer 3 Report

As it stands, the manuscript is pretty much unreadable due to the low quality English translation. First of all, the text must be written properly, as it is now, it hardly makes any sense. 

Author Response

Thank you for your time for reviewing. 
We already made modifications according the reviewers' suggestion.

Round 2

Reviewer 3 Report

With all due respect, I consider that vast language correction is needed before the manuscript can be published. 

Author Response

Thank you for your suggestion. This manuscript has been checked by the native English-speaking colleague.